# Learning Invariant Representations of Graph Neural Networks via Cluster Generalization

**Donglin Xia[1], Xiao Wang[2]\*, Nian Liu[1], Chuan Shi[1]\***
[1]Beijing University of Posts and Telecommunications
[2]Beihang University
{donglin.xia, nianliu, shichuan}@bupt.edu.cn, xiao_wang@buaa.edu.cn

## Abstract

Graph neural networks (GNNs) have become increasingly popular in modeling graph-structured data due to their ability to learn node representations by aggregating local structure information. However, it is widely acknowledged that the test graph structure may differ from the training graph structure, resulting in a structure shift. In this paper, we experimentally find that the performance of GNNs drops significantly when the structure shift happens, suggesting that the learned models may be biased towards specific structure patterns. To address this challenge, we propose the Cluster Information Transfer (**CIT**) mechanism[2], which can learn invariant representations for GNNs, thereby improving their generalization ability to various and unknown test graphs with structure shift. The CIT mechanism achieves this by combining different cluster information with the nodes while preserving their cluster-independent information. By generating nodes across different clusters, the mechanism significantly enhances the diversity of the nodes and helps GNNs learn the invariant representations. We provide a theoretical analysis of the CIT mechanism, showing that the impact of changing clusters during structure shift can be mitigated after transfer. Additionally, the proposed mechanism is a plug-in that can be easily used to improve existing GNNs. We comprehensively evaluate our proposed method on three typical structure shift scenarios, demonstrating its effectiveness in enhancing GNNs' performance.

## 1 Introduction

Graphs are often easily used to model individual properties and inter-individual relationships, which are ubiquitous in the real world, including social networks, e-commerce networks, citation networks. Recently, graph neural networks (GNNs), which are able to effectively employ deep learning on graphs to learn the node representations, have attracted considerable attention in dealing with graph data [13, 24, 32, 36, 9, 5]. So far, GNNs have been applied to various applications and achieved remarkable performance, *e.g.*, node classification [13, 32], link prediction [33, 14] and graph classification [6, 34].

Message-passing mechanism forms the basis of most graph neural networks (GNNs) [13, 24, 5, 9]. That is, the node representations are learned by aggregating feature information from the neighbors in each convolutional layer. So it can be seen that the trained GNNs are highly dependent on the local structure. However, it is well known that the graph structure in the real world always changes. For instance, the paper citations and areas would go through significant change as time goes by in citation network [11]. In social networks, nodes represent users and edges represent activity between users, because they will be changed dynamically, the test graph structure may also change [7]. This change in the graph structure is a form of distribution shift, which we refer to as graph structure shift. So

---

\*Corresponding authors.
[2]Code available at https://github.com/BUPT-GAMMA/CITGNN

the question naturally arises: *when the test graph structure shift happens, can GNNs still maintain stability in performance?*

We present experiments to explore this question. We generate graph structure and node features. Subsequently we train GNNs on the initially generated graph structure and gradually change the structures to test the generalization of GNNs (more details are in Section 2). Clearly, the performance consistently declines with changes (shown in Figure 1), implying that the trained GNNs are severely biased to one typical graph structure and cannot effectively address the structure shift problem.

This can also be considered as Out-Of-Distribution problem (OOD) on graph. To ensure good performance, most GNNs require the training and test graphs to have identically distributed data. However, this requirement cannot always hold in practice. Very recently, there are some works on graph OOD problem for node classification. One idea assumes knowledge of the graph generation process and derives regularization terms to extract hidden stable variables [28, 8]. However, these methods heavily rely on the graph generation process, which is unknown and may be very complex in reality. The other way involves sampling unbiased test data to make their distribution similar to that of the training data [37]. However, it requires sampling test data beforehand, which cannot be directly applied to the whole graph-level structure shift, *e.g.*, training the GNNs on a graph while the test graph is another new one, which is a very typical inductive learning scenario [24, 17].

Therefore, it is still technically challenging to learn the invariant representations which is robust to the structure shift for GNNs. Usually, the invariant information can be discovered from multiple structure environments [28], while we can only obtain one local structure environment given a graph. To avoid GNNs being biased to one structure pattern, directly changing the graph structure, *e.g.*, adding or removing edges, may create different environments. However, because the graph structure is very complex in the real world and the underlying data generation mechanism is usually unknown, it is very difficult to get the knowledge on how to generate new graph structures.

In this paper, we propose to learn the invariant representations of GNNs by transferring the cluster information of the nodes. First, we apply GNNs to learn the node representations, and then combine it with spectral clustering to obtain the cluster information in this graph. Here, we propose a novel Cluster Information Transfer (CIT) mechanism, because the cluster information usually captures the local properties of nodes and can be used to generate multiple local environments. Specifically, we characterize the cluster information using two statistics: the mean of cluster and the variance of cluster, and transfer the nodes to new clusters based on these two statistics while keeping the cluster-independent information. After training GNNs on these newly generated node representations, we aim to enhance their generalization ability across various test graph structures by improving the generalization ability on different clusters. Additionally, we provide insights into the transfer process from the embedding space and theoretically analyze the impact of changing clusters during structure shift can be mitigated after transfer. The contributions of our work are summarized as follows:

- We study the problem of structure shifts for GNNs, and propose a novel CIT mechanism to improve the generalization ability of GNNs. Our proposed mechanism is a friendly plug-in, which can be easily used to improve most of the current GNNs.

- Our proposed CIT mechanism enables that the cluster information can be transferred while preserving the cluster-independent information of the nodes, and we theoretically analyze that the impact of changing clusters during structure shift can be mitigated after transfer.

- We conduct extensive experiments on three typical structure shift tasks. The results well demonstrate that our proposed model consistently improves generalization ability of GNNs on structure shift.

## 2   Effect of structure distribution shift on GNN performance

In this section, we aim to explore the effect of structure shift on GNN performance. In the real world, the tendency of structure shift is usually unknown and complex, so we assume a relatively simple scenario to investigate this issue. For instance, we train the GNNs on a graph with two community structures, and then test the GNNs by gradually changing the graph structures. If the performance of GNNs drops sharply, it indicates that the trained GNNs are biased to the original graph, and cannot be well generalized to the new test graphs with structure shift.

We generate a network with 1000 nodes and divide all nodes into two categories on average, that is, 500 nodes are assigned label 0 and the other 500 nodes are assigned label 1. Meanwhile, node features, each of 50 dimensions, are generated by two Multivariate Gaussian distributions. The node features with same labels are generated by same Gaussian distribution. We employ the Stochastic Blockmodel (SBM) [12] to generate the graph structures. We set two community and set the generation edges probability on inter-community is 0.5% and intro-community is 0.05% . We randomly sample 20 nodes per class for training, and the rest are used for testing. Then we train GCN [13] and GAT [24] on this graph. The test graph structures are generated as follows: we decrease the inter-community edge probability from 0.5% to 0.25% and increase the intra-community probability from 0.05% to 0.3%.

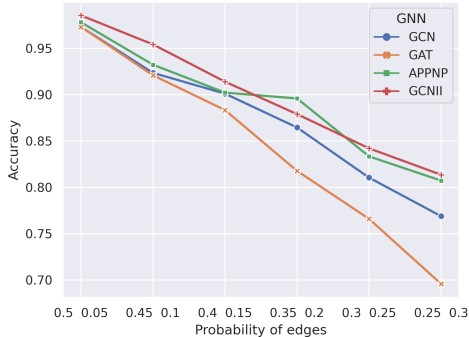

Figure 1: The node classification accuracy of GCN, GAT, APPNP and GCNII on generated data with structure shift. The x-axis is probability of edges (%).

The accuracy is shown in Figure 1. The x-axis is the probability of edges, where the first number is the edge probability of inter-community and the second number is the edge probability of intro-community. As we can see, because of the structure shift, the performance of GNN declines significantly. It shows that once the test graph pattern shifts, the reliability of GNNs becomes compromised.

## 3 Methods

**Notations:** Let $G = (\mathbf{A}, \mathbf{X})$ represent a training attributed graph, where $\mathbf{A} \in \mathbb{R}^{n \times n}$ is the symmetric adjacency matrix with $n$ nodes and $\mathbf{X} \in \mathbb{R}^{n \times d}$ is the node feature matrix, and $d$ is the dimension of node features. Specifically, $A_{ij} = 1$ represents there is an edge between nodes $i$ and $j$, otherwise, $A_{ij} = 0$. We suppose each node belongs to one of $C$ classes and focus on semi-supervised node classification. Considering that the graphs always change in reality, here we aim to learn the invariant representations for GNNs, where the overall framework is shown in Figure 2 and the whole algorithm is shown in A.1.

### 3.1 Clustering process

We first obtain the node representations through GNN layers:

$$\mathbf{Z}^{(l)} = \sigma(\tilde{\mathbf{D}}^{-1/2}\tilde{\mathbf{A}}\tilde{\mathbf{D}}^{-1/2}\mathbf{Z}^{(l-1)}\mathbf{W}_{GNN}^{(l-1)}), \tag{1}$$

where $\mathbf{Z}^{(l)}$ is the node representations from the $l$-th layer, $\mathbf{Z}^{(0)} = \mathbf{X}$, $\tilde{\mathbf{A}} = \mathbf{A} + \mathbf{I}$, $\tilde{\mathbf{D}}$ is the degree matrix of $\tilde{\mathbf{A}}$, $\sigma$ is a non-linear activation and $\mathbf{W}_{GNN}^{(l-1)}$ is the trainable parameters of GNNs. Eq. (1) implies that the node representations will aggregate the information from its neighbors in each layer, so the learned GNNs are essentially dependent on the local structure of each node. Apparently, if the structure shift happens in the test graph, the current GNNs may provide unsatisfactory results. Therefore, it is highly desired that the GNNs should learn the invariant representations while structure changing, so as to handle the structure shift in test graphs.

One alternative way is to simulate the structure change and then learn the invariant representations from different structures, which is similar as domain generalization [26, 35]. Motivated by this, we consider that the local properties of a node represents its domain information. Meanwhile, different clusters can capture different local properties in a graph, so we can consider cluster information as domain information of nodes. Based on this idea, we first aim to obtain the cluster information using spectral clustering [2]. Specifically, we compute the cluster assignment matrix $\mathbf{S}$ of node using a multi-layer perceptron (MLP) with softmax on the output layer:

$$\mathbf{S} = Softmax(\mathbf{W}_{MLP}\mathbf{Z}^{(l)} + \boldsymbol{b}), \tag{2}$$

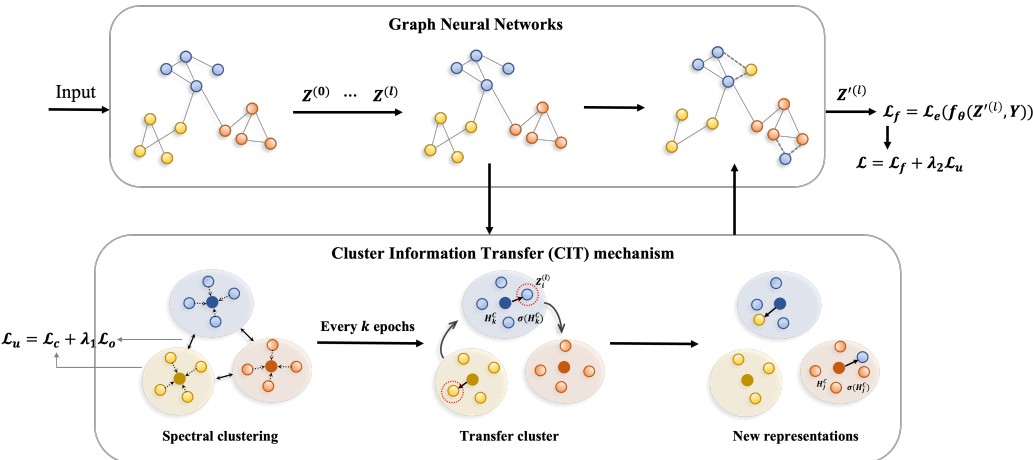

Figure 2: The overall framework of our proposed CIT mechanism on GNNs consists two parts: the traditional GNNs and Cluster Information Transfer (CIT) mechanism. After getting node representations from GNNs, we conduct CIT mechanism on node representations before the last layer of GNNs, which transfers the node to another cluster to generate new representations for training.

where $\mathbf{W}_{MLP}$ and $\boldsymbol{b}$ are trainable parameters of MLP. For assignment matrix $\mathbf{S} \in \mathbb{R}^{n \times m}$, and $m$ is the number of clusters. $s_{ij}$ represents the weight that node $i$ belongs to cluster $j$. The softmax activation of the MLP guarantees that the weight $s_{ij} \in [0, 1]$ and $\mathbf{S}^T \mathbf{1}_M = \mathbf{1}_N$.

For $\mathbf{S}$, we want to cluster strongly connected nodes. So we adopt the cut loss which evaluates the mincut given by $\mathbf{S}$:

$$\mathcal{L}_c = -\frac{Tr(\mathbf{S}^T \tilde{\mathbf{A}} \mathbf{S})}{Tr(\mathbf{S}^T \tilde{\mathbf{D}} \mathbf{S})}, \tag{3}$$

where $Tr$ is the trace of matrix. Minimizing $\mathcal{L}_c$ encourages the nodes which are strongly connected to be together.

However, directly minimizing $\mathcal{L}_c$ will make the cluster assignments are equal for all nodes, which means all nodes will be in the same cluster. So in order to avoid the clustering collapse, we can make the the clusters to be of similar size, and we use orthogonality loss to realize it:

$$\mathcal{L}_o = \left\| \frac{\mathbf{S}^T \mathbf{S}}{\|\mathbf{S}^T \mathbf{S}\|_F} - \frac{\mathbf{I}_M}{\sqrt{M}} \right\|_F, \tag{4}$$

where $\| \cdot \|_F$ indicates the Frobenius norm, $M$ is the number of clusters. Notably, when $\mathbf{S}^T \mathbf{S} = \mathbf{I}_M$, the orthogonality loss $\mathcal{L}_o$ will reach 0. Therefore, minimizing it can encourage the cluster assignments to be orthogonal and thus the clusters can be of similar size.

Overall, we optimize the clustering process by these two losses:

$$\mathcal{L}_u = \mathcal{L}_c + \lambda_1 \mathcal{L}_o, \tag{5}$$

where $\lambda_1$ is a coefficient to balance the mincut process and orthogonality process.

After obtaining the assignment matrix $\mathbf{S}$, we can calculate cluster representations $\mathbf{H}^c$ by average the node representations $\mathbf{Z}^{(l)}$:

$$\mathbf{H}^c = \mathbf{S}^T \mathbf{Z}^{(l)}. \tag{6}$$

## 3.2 Transfer cluster information

Now we obtain the cluster representations $\mathbf{H}^c$, and each cluster captures the information of local properties. Originally, given a graph, a node can be only in one domain, *i.e.*, one cluster. Next, we aim to transfer the node to different domains. Specifically, the cluster information can be characterized by two statistics, *i.e.*, the center of the cluster ($\mathbf{H}^c$) and the standard deviation of the cluster ($\sigma(\mathbf{H}^c_k)$):

$$\sigma(\mathbf{H}_k^c) = \sqrt{\Sigma_{i=1}^n s_{ik}(\mathbf{Z}_i^{(l)} - \mathbf{H}_k^c)^2}, \tag{7}$$

where $\mathbf{H}_k^c$ is the representation of the $k$-th cluster and $\mathbf{Z}_i^{(l)}$ is the representation of node $i$.

Then we can transfer the node $i$ in the $k$-th cluster to the $j$-th cluster as follows:

$$\mathbf{Z}_i'^{(l)} = \sigma(\mathbf{H}_j^c)\frac{\mathbf{Z}_i^{(l)} - \mathbf{H}_k^c}{\sigma(\mathbf{H}_k^c)} + \mathbf{H}_j^c, \tag{8}$$

where $k$ is the cluster that node $i$ belongs to, and $j$ is the cluster which is randomly selected from the remaining clusters.

Here, we explain the Eq. (8) in more details. As shown in Figure 3, the center of cluster $k$ is $\mathbf{H}_k^c$, and node $i$ belongs to this cluster, where the representation of node $i$ is $\mathbf{Z}_i^{(l)}$. It can be seen that $\mathbf{H}_k^c$ is obtained by aggregating and averaging the local structure information, which captures the cluster information. Therefore, $\mathbf{Z}_i^{(l)} - \mathbf{H}_k^c$ represents the cluster-independent information. Then, the CIT mechanism can be seen as we transfer the node to a new position from embedding space. The standard deviation is the weighted distance of nodes from the center, which is the aggregating scope of the clusters. After the transfer, the target node surrounds a new cluster with new cluster information, while keeping the cluster-independent information.

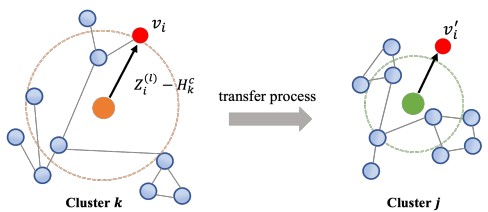

Figure 3: We show two parts of one graph. The orange and green points represent two clusters in one graph. The circle is aggregating scope of cluster. And the red point represents the target node we transfer from orange cluster to green cluster.

The above process only transfers the nodes on the original domain. Moreover, in order to improve the robustness of the model for unknown domain, we increase the uncertainty and diversity of model [16, 21]. Based on this, we add Gaussian perturbations to this process. The whole transfer becomes:

$$\mathbf{Z}_i'^{(l)} = (\sigma(\mathbf{H}_j^c) + \epsilon_\sigma \Sigma_\sigma)\frac{\mathbf{Z}_i^{(l)} - \mathbf{H}_k^c}{\sigma(\mathbf{H}_k^c)} + (\mathbf{H}_j^c + \epsilon_\mu \Sigma_\mu). \tag{9}$$

The statistics of Gaussian is determined by the whole features:

$$\epsilon_\sigma \sim \mathcal{N}(0,1), \epsilon_\mu \sim \mathcal{N}(0,1), \tag{10}$$

$$\Sigma_\sigma^2 = \sigma(\sigma(\mathbf{H}^c)^2)^2, \Sigma_\mu^2 = \sigma(\mathbf{H}^c)^2. \tag{11}$$

Here, with the Gaussian perturbations, we can generate new clusters based on the original one and the nodes can be further transferred to more diverse domains.

The proposed transfer process offers a solution to overcome the limitations of changing graph structures when generating nodes in different domains. Traditional approaches require changing the graph structure by adding or deleting edges, which can be challenging due to the lack of prior knowledge on how the changes in graph structure may affect the domain. Here, our proposed CIT mechanism addresses this challenge by directly transferring a node to another domain through the manipulation of cluster properties of nodes in the embedding space. The CIT method is implemented before the classifier in GNNs training, making it backbone agnostic and compatible with any GNN last layer.

### 3.3 Objective function

With CIT mechanism, we have the generated representation $\mathbf{Z}_i'^{(l)}$ in other clusters for node $i$. In traditional GNNs training, we input the representation $\mathbf{Z}^{(l)}$, where the $i$-th row vector represents the representation of node $i$, to the cross-entropy loss [13]. Here, we randomly select a part of nodes, and replace original representation $\mathbf{Z}_i^{(l)}$ with the generated representation $\mathbf{Z}_i'^{(l)}$. Then we can obtain the new representation matrix $\mathbf{Z}'^{(l)}$, and optimize the cross-entropy loss based on it as follows:

$$\mathcal{L}_f = \mathcal{L}_e(f_\theta(\mathbf{Z}'^{(l)}), \mathbf{Y}), \tag{12}$$

where $\mathcal{L}_e$ is cross-entropy loss function, $f_\theta$ is the classifier, and $\mathbf{Y}$ is the labels.

Finally, the overall optimization function is as follows:

$$\mathcal{L} = \mathcal{L}_f + \lambda_2 \mathcal{L}_u, \tag{13}$$

where $\lambda_2$ is a coefficient to balance the classification process and clustering process. In the optimization process, we randomly select $n \times p$ nodes and then conduct Eq. (9) every $k$ epochs.

### 3.4 Theoretical analysis

In this section, we theoretically analyze our CIT mechanism from the perspective of domain adoption theory. Since the Eq. (8) is the cornerstone of our CIT mechanism and Eq. (9) is an extending from it, we use Eq. (8) to represent our CIT mechanism in the following proof for convenience.

We analyze that the classifier using the new generated representation $\mathbf{Z}'^{(l)}$ in Eq. (8) has better generalization on the structure shift. As mentioned before, cluster information can capture local properties of nodes, so we convert this problem to analyzing the classifier has better generalization on cluster shift. To achieve this, following [31], we take the binary classification task as the example, and analyze the relationship between the decision boundary with cluster information. For analysis, we follow [31] using a fisher classifier and analyse the relationship between them. According to Fisher's linear discriminant analysis [1, 23, 3], the decision boundary of fisher classifier depends on two statistics $Var(Z)$ and $Cov(Z, Y)$.

**Theorem 1.** *The decision boundary of fisher classifier is affected by the cluster information.*

The proof is given in Appendix A.3, where we calculate the mathematical expression of the classification boundary. Theorem 1 indicates that the cluster information will "help" the classifier to make decisions. But when the graph structure changes, the cluster information of the nodes also changes. Therefore, the classifier using cluster information to make decisions is not reliable.

**Theorem 2.** *Let $Z_R$ represent the node representations in cluster $R$. Assume that there are $p$ percent of nodes are transferred from cluster $R$ to cluster $D$ by $\Sigma_D \frac{Z_R - \mu_R}{\Sigma_R} + \mu_D$. After the transfer, the impact of changing clusters during structure shift can be mitigated.*

The proof is given in Appendix A.4. Comparing the expression of classifier before and after our CIT mechanism, we can find that we mitigate the impact of changing clusters during structure shift, enhancing the robustness of the model against such changes.

## 4 Experiment

**Datasets and baselines.** To comprehensively evaluate the proposed CIT mechanism, we use six diverse graph datasets. Cora, Citeseer, Pubmed [20], ACM, IMDB [27] and Twitch-Explicit [17]. Details of datasets are in Appendix B.1. Our CIT mechanism can be used for any other GNN backbones. We plug it in four well-known graph neural network methods, GCN [13], GAT [24], APPNP [9] and GCNII [5]. Meanwhile, we compare it with two graph OOD methods which are also used for node-classification, SR-GNN [37] and EERM [28]. We combine them with four methods mentioned before. Meanwhile, we use CIT-GNN(w/o) to indicate that no Gaussian perturbations are added.

**Experimental setup.** For Cora, Citeseer and Pubmed, we construct structure shift by making perturbations on the original graph structure. For ACM and IMDB, we test them using two relation structures. For Twitch-Explicit, we use six different networks. We take one network to train, one network to validate and the rest of networks to test. So we divide these datasets into three categories: perturbation on graph structures, Multiplex networks and Multigraph. The implementation details are given in Appendix B.1.

### 4.1 Perturbation on graph structures data

In this task, we use Cora, Citeseer and Pubmed datasets. We train the model on original graph. We create new graph structures by randomly adding 50%, 75% and deleting 20%, 50% edges of original graph. To comprehensively evaluate our model on random structure shift, we test our model on the

Table 1: Quantitative results ($\%\pm\sigma$) on node classification for perturbation on graph structures data while the superscript refers to the results of paired t-test between original model and CIT-GNN (* for 0.05 level and ** for 0.01 level).

| Method | ADD-0.5 | | | | | |
| | Cora | | Citeseer | | Pubmed | |
| | Acc | Macro-f1 | Acc | Macro-f1 | Acc | Macro-f1 |
|---|---|---|---|---|---|---|
| GCN | 76.12±0.91 | 74.95±0.88 | 65.43±0.68 | 63.02±0.59 | 72.58±1.10 | 71.84±1.00 |
| SR-GCN | 75.70±0.90 | 74.35±0.85 | 66.03±1.30 | 62.67±0.80 | 72.15±1.80 | 70.63±1.90 |
| EERM-GCN | 75.33±0.87 | 74.39±0.89 | 64.05±0.49 | 60.97±0.61 | -- | |
| CIT-GCN(w/o) | 76.88±0.34 | 75.63±0.47 | 66.67±0.55 | 64.21±0.47 | 72.83±0.32 | 72.01±0.21 |
| CIT-GCN | **76.98±0.49*** | **75.88±0.44*** | **67.65±0.44**** | **64.42±0.10**** | **73.76±0.40*** | **72.94±0.30*** |
| GAT | 77.04±0.30 | 76.15±0.40 | 64.42±0.41 | 61.74±0.30 | 71.30±0.52 | 70.80±0.43 |
| SR-GAT | 77.35±0.75 | 76.49±0.77 | 64.80±0.29 | 61.98±0.11 | 71.55±0.52 | 70.79±0.64 |
| EERM-GAT | 76.15±0.38 | 75.32±0.29 | 62.05±0.79 | 59.01±0.65 | -- | |
| CIT-GAT(w/o) | **77.37±0.57** | **76.73±0.47** | 65.23±0.58 | **63.30±0.67**** | 71.92±0.68 | 71.12±0.56 |
| CIT-GAT | 77.23±0.42 | 76.26±0.28 | **66.33±0.24**** | 63.07±0.37 | **72.50±0.74** | **71.57±0.82** |
| APPNP | 79.54±0.50 | 77.69±0.70 | 66.96±0.76 | 64.08±0.66 | 75.88±0.81 | 75.37±0.66 |
| SR-APPNP | 80.00±0.70 | 78.56±0.87 | 65.20±0.23 | 62.77±0.36 | 75.85±0.55 | 75.43±0.58 |
| EERM-APPNP | 78.10±0.73 | 76.72±0.69 | 66.30±0.91 | 63.08±0.77 | -- | |
| CIT-APPNP(w/o) | 79.79±0.40 | 77.95±0.32 | 68.06±0.52 | 65.30±0.32 | 76.21±0.55 | 75.23±0.37 |
| CIT-APPNP | **80.50±0.39*** | **78.86±0.24*** | **68.54±0.71*** | **65.51±0.45*** | **76.64±0.40*** | **75.89±0.48*** |
| GCNII | 76.98±0.92 | 74.92±0.97 | 63.16±1.20 | 61.14±0.78 | 74.03±1.10 | 73.37±0.75 |
| SR-GCNII | 77.55±0.21 | 75.09±0.41 | 64.74±1.86 | 62.44±1.53 | 75.10±0.78 | 74.36±0.95 |
| EERM-GCNII | **79.05±1.10** | **76.62±1.23** | 65.10±0.64 | 62.02±0.76 | -- | |
| CIT-GCNII(w/o) | 77.64±0.63 | 75.22±0.61 | 65.87±0.80 | **63.36±0.75**** | 75.00±0.47 | 74.70±0.54 |
| CIT-GCNII | 78.28±0.88 | 75.82±0.73 | **66.12±0.97**** | 63.17±0.85 | **75.95±0.63*** | **75.47±0.76*** |
| | ADD-0.75 | | | | | |
| GCN | 72.37±0.55 | 71.09±0.36 | 63.34±0.60 | 61.09±0.54 | 72.48±0.31 | 71.06±0.58 |
| SR-GCN | 72.70±1.10 | 72.19±1.20 | 62.72±1.80 | 59.58±2.10 | 70.35±2.10 | 69.14±2.30 |
| EERM-GCN | 72.30±0.21 | 71.68±0.47 | 61.65±0.54 | 58.55±0.68 | -- | |
| CIT-GCN(w/o) | 72.90±0.53 | 71.70±0.67 | 64.83±0.79 | 62.33±0.56 | 73.00±0.46 | 72.30±0.56 |
| CIT-GCN | **74.44±0.75*** | **73.37±0.86*** | **64.80±0.65*** | **62.52±0.46*** | **73.20±0.36*** | **72.33±0.41*** |
| GAT | 73.86±0.45 | 72.79±0.24 | 63.42±1.00 | 61.35±0.79 | 70.88±0.67 | 69.97±0.64 |
| SR-GAT | 74.28±0.28 | 73.46±0.43 | 64.27±1.10 | 62.04±0.90 | 70.36±0.69 | 69.24±0.88 |
| EERM-GAT | 72.62±0.43 | 72.28±0.30 | 62.00±0.43 | 60.30±0.50 | -- | |
| CIT-GAT(w/o) | 74.64±0.77 | 73.76±0.87 | 63.83±0.79 | 62.33±0.56 | 71.00±0.77 | 69.30±0.56 |
| CIT-GAT | **74.75±0.40*** | **73.67±0.58*** | **64.74±0.67*** | **62.23±0.77*** | **71.90±0.89** | **70.88±0.67** |
| APPNP | 75.86±0.84 | 73.97±0.87 | 65.71±1.01 | 63.65±0.86 | 74.53±0.77 | 73.99±0.80 |
| SR-APPNP | 76.00±0.30 | 73.98±0.40 | 64.80±0.37 | 62.78±0.21 | 75.40±0.58 | 74.30±0.68 |
| EERM-APPNP | 75.30±0.65 | 74.87±0.62 | 64.90±0.69 | 62.32±0.60 | -- | |
| CIT-APPNP(w/o) | 77.53±0.67 | 75.66±0.54 | 67.01±0.81 | 64.78±0.73 | **76.05±0.87*** | **75.91±0.92**** |
| CIT-APPNP | **78.02±0.56**** | **76.53±0.78**** | **66.06±0.95*** | **63.81±0.58*** | 75.70±0.41 | 75.88±0.83 |
| GCNII | 73.16±1.05 | 71.01±1.39 | 62.48±1.20 | 60.80±0.40 | 75.78±0.58 | 75.15±0.61 |
| SR-GCNII | 75.03±0.50 | 72.28±0.93 | 60.90±2.10 | 59.00±1.80 | 75.98±1.10 | 75.79±0.90 |
| EERM-GCNII | 75.50±1.20 | 73.60±1.30 | 62.40±0.97 | 59.11±0.81 | -- | |
| CIT-GCNII(w/o) | 74.64±0.63 | 72.84±0.87 | 64.58±0.87 | 62.47±0.69 | 75.78±0.99 | 74.20±1.10 |
| CIT-GCNII | **77.08±1.22**** | **75.15±1.45**** | **65.82±1.04**** | **63.27±0.73**** | **76.13±1.12** | **75.99±1.17** |

new graphs. We follow the original node-classification settings [13] and use the common evaluation metrics, including Macro-F1 and classification accuracy. For brief presentation, we show results of deleting edges in Appendix B.2.

The results are reported in Table 1. From the table we can see that the proposed CIT-GNN generally achieves the best performance in most cases. Especially, for Acc and Macro-f1, our CIT-GNN achieves maximum relative improvements of 5.35% and 4.1% respectively on Citesser-Add-0.75. The results demonstrate the effectiveness of our CIT-GNN. We can also see that CIT-GNN improves the four basic methods, so the results show that our method can improve the generalization ability of the basic models. Meanwhile, our mechanism is to operate the node representation at the embedding level, which can be used for any GNN backbones.

## 4.2 Multiplex networks data

In this task, we use Multiplex networks datasets ACM and IMDB to evaluate the capabilities of our method on different relational scenarios. Both of them have two relation structures. We construct structure shift by taking the structure of one relation for training and the other for testing respectively. The partition of data follows [27]. The results are reported in Table 2 and the relation structure shown in the table is the training structure. Different from the first experiment, the structure shift is not random, which is more intense because the new graph is not based on the original graph. As can be seen, our proposed CIT-GNN improves the four basic methods in most cases, and outperforms

Table 2: Quantitative results (%±$\sigma$) on node classification for multiplex networks data and the relation in table is for training, while the superscript refers to the results of paired t-test between original model and CIT-GNN (* for 0.05 level and ** for 0.01 level).

| Method | ACM | | | | IMDB | | | |
| --- | --- | --- | --- | --- | --- | --- | --- | --- |
| | PAP | | PLP | | MDM | | MAM | |
| | Acc | Macro-f1 | Acc | Macro-f1 | Acc | Macro-f1 | Acc | Macro-f1 |
| GCN | 64.65±1.91 | 60.66±1.88 | 80.26±1.98 | 79.71±1.94 | 52.36±1.40 | 48.55±1.60 | 58.98±1.11 | 57.01±1.72 |
| SR-GCN | 67.75±1.20 | 68.51±1.10 | 82.14±2.10 | 81.88±2.32 | 51.94±0.97 | 50.76±0.85 | **59.84±1.08** | **58.21±1.11** |
| EERM-GCN | 66.85±1.87 | 67.84±1.54 | 82.16±1.87 | 82.16±1.67 | 54.07±1.83 | 51.80±1.62 | 57.21±1.93 | 56.52±1.66 |
| CIT-GCN(w/o) | 67.53±1.52 | 65.32±1.93 | 81.30±1.58 | 80.98±1.82 | 53.67±1.79 | 50.71±1.56 | 57.93±1.32 | 56.23±1.24 |
| CIT-GCN | **68.06±1.13**\*\* | **68.79±1.27**\*\* | **82.67±1.55**\* | **82.56±1.66**\* | **55.42±1.88**\*\* | **52.75±1.65**\*\* | 56.68±1.45 | 54.66±1.97 |
| GAT | 66.35±1.81 | 64.23±2.25 | 82.48±1.73 | 82.50±1.65 | 51.59±1.13 | 48.26±1.27 | 58.64±1.87 | 57.72±1.94 |
| SR-GAT | 67.20±1.87 | 67.89±2.13 | 84.61±1.34 | 84.49±1.65 | 50.81±1.92 | 46.62±1.73 | 59.05±1.57 | 57.42±1.84 |
| EERM-GAT | 67.67±1.17 | **68.22±1.25** | 79.25±1.27 | 78.84±0.99 | 52.24±1.28 | 50.86±1.35 | 58.20±1.65 | 57.25±1.59 |
| CIT-GAT(w/o) | 67.15±1.23 | 67.34±1.43 | 83.45±1.54 | 83.01±1.46 | **53.91±0.96**\*\* | **51.90±1.21**\* | 57.18±1.55 | 56.82±1.23 |
| CIT-GAT | **68.49±1.32**\* | 68.15±1.39\*\* | **85.75±1.76**\*\* | **85.64±1.32**\*\* | 52.86±0.98 | 51.06±1.01 | **59.51±1.73** | **58.40±1.46** |
| APPNP | 78.49±1.33 | 79.00±1.56 | 86.76±1.22 | 86.74±1.85 | 51.78±1.01 | 46.57±1.32 | 62.01±1.21 | 61.34±1.56 |
| SR-APPNP | 77.60±1.28 | 76.25±1.77 | 86.16±1.37 | 86.16±1.52 | 55.02±1.98 | 51.74±2.03 | 60.70±1.08 | 60.14±1.32 |
| EERM-APPNP | 80.89±1.82 | 80.43±1.65 | 83.58±1.58 | 83.46±1.84 | 54.32±0.96 | 51.03±1.07 | 61.27±1.75 | 60.30±1.87 |
| CIT-APPNP(w/o) | 81.66±1.12 | 81.35±1.01 | 86.60±0.98 | 86.52±0.87 | **56.37±1.41**\*\* | **54.13±1.78**\*\* | 61.81±0.92 | 60.98±1.10 |
| CIT-APPNP | **81.70±1.58**\*\* | **81.60±1.47**\* | **87.19±1.21**\* | **87.16±1.02**\* | 55.86±1.54 | 52.32±1.73 | **62.50±1.45** | **62.00±1.54** |
| GCNII | 77.92±1.64 | 76.73±1.77 | 81.88±1.05 | 81.21±1.32 | 52.71±1.86 | 47.08±1.77 | 52.45±2.01 | 47.61±2.35 |
| SR-GCNII | 78.91±1.73 | 78.77±1.76 | 84.32±0.89 | 83.22±1.01 | 53.52±1.56 | 49.87±1.36 | 54.20±2.33 | 49.00±1.98 |
| EERM-GCNII | 78.82±1.23 | 79.24±1.76 | 83.81±1.21 | 83.61±1.02 | 53.56±1.23 | 50.27±1.42 | **54.32±1.98** | **49.70±1.87** |
| CIT-GCNII(w/o) | 78.10±1.54 | 78.80±1.64 | 84.85±1.41 | 84.62±1.54 | 53.42±2.01 | 49.10±1.58 | 52.30±1.87 | 48.01±2.10 |
| CIT-GCNII | **79.73±1.61**\* | **79.26±1.32**\*\* | **85.23±1.93**\*\* | **85.06±1.89**\*\* | **54.20±1.88**\* | **50.91±1.95**\* | 53.69±1.89 | 48.84±2.01 |

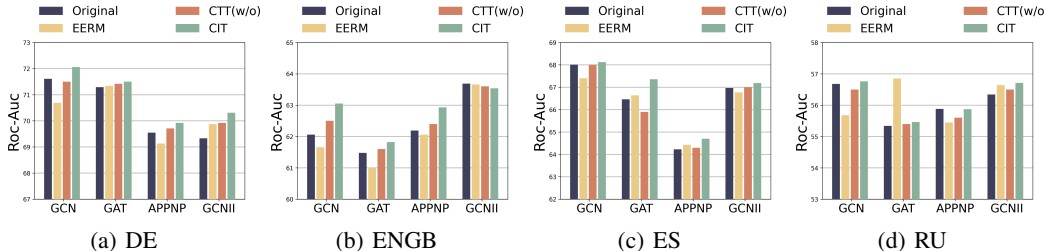

(a) DE     (b) ENGB     (c) ES     (d) RU

Figure 4: ROC-AUC on Twitch where we compare different GNN backbones.

SR-GNN and EERM, implying that our CIT-GNN can improve the generalization ability of the basic models.

## 4.3 Multigraph data

In this task, we use Multigraph dataset Twitch-Explicit, and its each graph is collected from a particular region. In this dataset, the node features also change but they still share the same input feature space and output space. The graphs are collected from different regions. So different structures are determined by different regions. To comprehensively evaluate our method, we take the FR for training, TW for validation, and test our model on the remaining four graphs (DE, ENGB, RS, EU). We choose ROC-AUC score for evaluation because it is a binary classification task. Since SR-GNN is not suitable for this dataset, only the base model and EERM are compared. The results are reported in Figure 4. Comparing the results on four graph, our model makes improvement in most cases, which verifies the effectiveness about generalizing on multigraph scenario.

## 4.4 Analysis of hyper-parameters

**Analysis of $p$.** The probability of transfer $p$ determines how many nodes to transfer every time. It indicates the magnitude of the CIT mechanism. We vary its value and the corresponding results are shown in Figure 5. With the increase of $p$, the performance goes up first and then declines, which indicates the performance benefits from an applicable selection of $p$.

**Analysis of $k$.** We make transfer process every $k$ epochs. We vary $k$ from 1 to 50 and and plot the results in Figure 6. Zero represents the original model accuracy without out CIT mechanism. Notably, the performance remains consistently stable across the varying values of $k$, indicating the robustness of the model to this parameter.

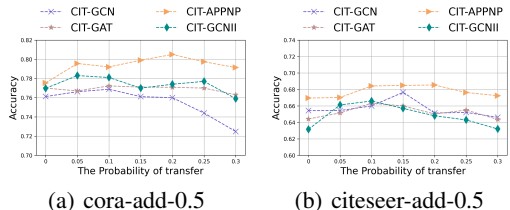

(a) cora-add-0.5      (b) citeseer-add-0.5

Figure 5: Analysis of the probability of transfer.

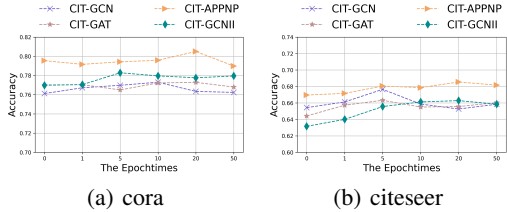

(a) cora             (b) citeseer

Figure 6: Analysis of the epochtimes.

**Analysis of** $m$. The number of clusters is the most important parameter in the clustering process. We choose silhouette coefficient to measure the clustering performance and vary $m$ from 20 to 100 unevenly. Then we calculate the accuracy and Silhouette Coefficient. The corresponding results are shown in Figure 7. As we can see, accuracy changes synchronously with silhouette coefficient. We infer that the performance of our model is related to the clustering situation, and when the clustering process performs well, our model also performs well.

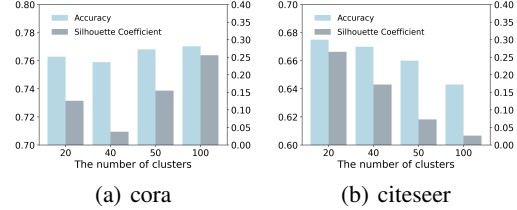

(a) cora             (b) citeseer

Figure 7: The number of clusters is varying on cora and citeseer. The accuracy corresponds to the right vertical axis, while the Silhouette Coefficient values correspond to the left vertical axis.

## 5 Related work

**Graph neural networks.** Recently, Graph Neural Networks have been widely studied. GCN [13] proposes to aggregate the node features from the one-hop neighbors. GAT [24] designs an attention mechanism to aggregate node features from neighbors. PPNP [9] utilizes PageRank's node propagation way to aggregate node features and proposes an approximate version. GCNII [5] extends GCN by introducing two effective techniques: initial residual and identity mapping, which make the network deeper. This is a fast growing research field, and more detailed works can be found in [4, 22].

**OOD generalization of GNNs.** Out-Of-Distribution (OOD) on graph has attracted considerable attention from different perspectives. For node-classification, [8] shows node selection bias drastically affects the performance of GNNs and investigates it from point view of causal theory. [37] explores the invariant relationship between nodes and proposes a framework to account for distributional differences between biased training data. [28] handles it by minimizing the mean and variance of risks from multiple environments which are generated by adversarial context generators. For graph classification, [15] proposes to capture the invariant relationships between predictive graph structural information and labels in a mixture of latent environments. [29, 18] find invariant subgraph structures from a causal perspective to improve the generalization ability of graph neural networks. [10] builds a graph OOD bench mark including node-level and graph-level methods and two kinds of distribution shift which are covariate shift and concept shift.

**Graph clustering with graph neural networks.** As graph neural networks continue to perform better in modeling graph data, some GNN-based graph clustering methods have been widely applied. Deep attentional embedded graph clustering [25] uses an attention network to capture the importance of the neighboring nodes and employs the KL-divergence loss in the process of graph clustering. [2] achieves the graph clustering process by obtaining the assignment matrix through minimizing optimizing its spectral objective.[19] uses a new objective function for clustering combining graph spectral modularity maximization and a new regularization method.

## 6 Conclusion

In this paper, we explore the impact of structure shift on GNN performance and propose a CIT mechanism to help GNNs learn invariant representations under structure shifts. We theoretically analyze that the impact of changing clusters during structure shift can be mitigated after transfer.

Moreover, the CIT mechanism is a friendly plug-in, and the comprehensive experiments well demonstrate the effectiveness on different structure shift scenarios.

**Limitations and broader impact.** One potential limitation lies in its primary focus on node-level tasks, while further investigation is needed to explore graph-level tasks. Although our CIT mechanism demonstrates significant advancements, certain theoretical foundations remain to be fully developed. Our work explores the graph from the perspective of the embedding space, thereby surpassing the limitations imposed by graph topology, and offers a fresh outlook on graph analysis.

## Acknowledgments and Disclosure of Funding

This work is supported in part by the National Natural Science Foundation of China (No. U20B2045, 62192784, U22B2038, 62002029, 62172052, 62322203).

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
