**Algorithm 1:** GNNs with the CIT mechanism

---

**Input** : Graph $G = (\mathbf{A}, \mathbf{X})$, label $\mathbf{Y}$
**Params** : the probability of transfer $p$, the epochtimes $k$,
  the number of clusters $m$, total iterations $T$
**Initialize** : GNN model $f_{GNN}$,
  classifier $f_\theta$ (usually the last layer of GNN)
**Output** : GNN model $f_{GNN}$, classifier $f_\theta$

---

**1 for** $epoch = 1 \ to \ T$ **do**
**2**  Node representation $\mathbf{Z}^{(l)}$ from Eq. (1)
**3**  Cluster representation $\mathbf{H}^c$ from Eq. (2) and Eq. (6)
**4**  Clustering loss $\mathcal{L}_u$ from Eq. (5)
**5**  **if** $epoch \ \% \ k == 0$ **then**
**6**    Randomly sample $n \times p$ nodes to calculate Eq. (9)
**7**    Get the new representation $\mathbf{Z}'^{(l)}$
**8**  **else**
**9**    Keep the node representation $\mathbf{Z}'^{(l)} \leftarrow \mathbf{Z}^{(l)}$
**10**  **end**
**11**  Classification loss $\mathcal{L}_f$ from Eq. (12)
**12**  Update $f_{GNN}, f_\theta$ with Eq. (13)
**13 end**

---

# A  More details of Section 3

## A.1  Three-Fold optimization

In this section, we detail the process of our CIT mechanism in three-fold optimization, shown in Algorithm1.

## A.2  Computational complexity

Our CIT mechanism has two parts of computation: Clustering process and Cluster Information Transfer process. Let $N$ represent the number of nodes, and $K$ represent the number of clusters. The computational complexity of clustering process is $\mathcal{O}(N^2K + NK^2) = \mathcal{O}(NK(N + K))$. Since the adjacency matrix is usually sparse, the computational complexity can be reduced to $\mathcal{O}(EK)$, where $E$ is the number of non-zero edges in the adjacency matrix. The computational complexity of Cluster Information Transfer process is $\mathcal{O}(pN)$, where $p$ is the probability of transfer. So the computational complexity is $\mathcal{O}(K(E + NK) + pN)$. The space complexity which depends on the dimension of the assignment matrix is $\mathcal{O}(NK)$.

## A.3  Proof of Theorem1

Specifically, we simplify $\mathbf{Z}^{(l)}$ in Eq. (1) as $\mathbf{Z}$, and assume that label $Y \in \{0, 1\}$. There are two clusters $e \in \{D, R\}$. We give the statistics of data. The mean of node representations in cluster $D$ is $E(Z|e = D) = \mu_D$, and the variance of node representations in cluster $D$ is $Var(Z|e = D) = \Sigma_D^2$. Similarly, $E(Z|e = R) = \mu_D$, $Var(Z|e = R) = \Sigma_R^2$. We define the probability of label 0 as $\pi_0 = P(Y = 0)$, label 1 as $\pi_1 = P(Y = 1)$, and then the cluster probability $\pi_D = P(e = D)$ and $\pi_R = P(e = R)$. The conditional probability of label given cluster $D$ as $\pi_{0|D} = P(Y = 0|e = D)$, $\pi_{1|D} = P(Y = 1|e = D)$ and the conditional probability of label given cluster $R$ as $\pi_{0|R} = P(Y = 0|e = R)$, $\pi_{1|R} = P(Y = 1|e = R)$. For analysis, we use $E(Z|Y = 1) = \mu_1$ to represent the mean of node representations with label 1, and $E(Z|Y = 0) = \mu_0$ to represent the mean of node representations with label 0. We assume that there is statistic spurious correlation between clusters and labels, *i.e.*, all label information can be obtained through the label information in each cluster as $\frac{\mu_D}{\pi_{1|D}} + \frac{\mu_R}{\pi_{1|R}} = \mu_1$. At first, we calculate the form of the decision boundary through the data statistics given above and find that the label distribution in clusters affects the decision boundary. And then, we

make our transfer on the original data and find that the label distribution in clusters has less influence on it.

*Proof.* Firstly, we use the statistics of node representations in different clusters and clusters probability to calculate the variance:

$$
\begin{aligned}
Var(Z) &= E(Z^2|e=D)\pi_D + E(Z^2|e=R)\pi_R - E(Z)^2 \\
&= (Var(Z|e=D) + E(Z|e=D)^2)\pi_D \\
&\quad + (Var(Z|e=R) + E(Z|e=R)^2)\pi_R - E(Z)^2 \\
&= (\Sigma_D^2 + \mu_D^2)\pi_D + (\Sigma_R^2 + \mu_R^2)\pi_R - (\mu_D\pi_D + \mu_R\pi_R)^2.
\end{aligned}
\tag{14}
$$

Then we calculate the covariance of $Z$ and $Y$ based on the correlation assumption:

$$
\begin{aligned}
Cov(Z,Y) &= E(ZY) - E(Y)E(Z) - E(Y)E(Z) + E(Z)E(Y) \\
&= E[E(ZY|Y) - YE(Z) - E(Y)E(Z|Y) + E(X)E(Y)] \\
&= E[(E(Z|Y) - E(E(Z|Y)))(Y - E(Y))] \\
&= Cov((\frac{\mu_D}{\pi_{1|D}} + \frac{\mu_R}{\pi_{1|R}} - \frac{\mu_D}{\pi_{0|D}} - \frac{\mu_R}{\pi_{0|R}})Y, Y) \\
&= (\frac{\mu_D}{\pi_{1|D}} + \frac{\mu_R}{\pi_{1|R}} - \frac{\mu_D}{\pi_{0|D}} - \frac{\mu_R}{\pi_{0|R}})\pi_0\pi_1.
\end{aligned}
\tag{15}
$$

Combining Eq. (14) and Eq. (15) we can see that, the label distribution in cluster $\pi_{Y|e}$ affects the covariance $Cov(Z,Y)$. So in this case, the decision boundary is directly influenced by cluster information. $\square$

## A.4 Proof of Theorem2

*Proof.* For analysis, we assume that there are $n_D$ nodes belonging to cluster $D$ and $n_R$ nodes belonging to cluster $R$. So after the transfer, the probability of cluster $D$ is $\pi'_D = \frac{n_D + n_R p}{n_D + n_R}$ and probability of cluster $R$ is $\pi'_R = \frac{n_R - n_R p}{n_D + n_R}$. The new variance can be calculated as follows:

$$
\begin{aligned}
Var(Z) &= (\Sigma_D'^2 + \mu_D^2)\pi'_D + (\Sigma_R'^2 + \mu_R^2)\pi'_R - (\mu_D\pi'_D + \mu_R\pi'_R)^2 \\
&= (\frac{\Sigma_D^2 n_D}{n_D + n_R p} + \frac{n_D}{n_D + n_R p}\mu_D^2 + \frac{n_R p}{n_D + n_R p}\mu_R^2 \\
&\quad - (\frac{n_D}{n_D + n_R p}\mu_D + \frac{n_R p}{n_D + n_R p}\mu_R)^2 + \mu_D^2)(\frac{n_D + n_R p}{n_D + n_R}) \\
&\quad + (\Sigma_R^2 + \mu_R^2)\frac{n_R - n_R p}{n_D + n_R} - (\mu_D\frac{n_D + n_R p}{n_D + n_R} + \mu_R\frac{n_R - n_R p}{n_D + n_R})^2.
\end{aligned}
\tag{16}
$$

We use $\pi'_{Y|e}$ to represent new label distribution in each clusters. Then the new covariance can be represented as follows:

$$
Cov(Z,Y) = (\frac{\mu_D}{\pi'_{1|D}} + \frac{\mu_R}{\pi'_{1|R}} - \frac{\mu_D}{\pi'_{0|D}} - \frac{\mu_R}{\pi'_{0|R}})\pi_0\pi_1.
\tag{17}
$$

From Eq. (16) and Eq. (17) we can see, the label distribution in cluster still have no effect on $Var(Z)$. So we analyze the $\pi'_{Y|e}$ which affects the $Cov(Z,Y)$. We take cluster $D$ as an example. We use $n_{D0}$ and $n_{R0}$ to represent the number of nodes with label 0 in each cluster. Similarly, $n_{D1}$ and $n_{R1}$ to represent the number of nodes with label 1 in each cluster. After the transfer, we calculate the probability of new label-0 in cluster $\pi'_{0|D} = \frac{n_{D0} + pn_R\pi_{0|R}}{n_D + n_R p} = \frac{n_{D0} + pn_{R0}}{n_D + n_R p}$. We can see that the conditional probability $\pi'_{0|D}$ approaches to $\pi_0$, which is as same as label 1, meaning that the effect

Table 3: Data statistics.

| Datasets | Nodes | Edges | Features | Classes | Structures |
|---|---|---|---|---|---|
| Cora | 2708 | 5429 | 1433 | 7 | 1 |
| Citeseer | 3327 | 4732 | 3703 | 6 | 1 |
| Pubmed | 19717 | 44324 | 500 | 3 | 1 |
| ACM | 3025 | 29281 | 1830 | 3 | PAP |
|  |  | 2210761 |  |  | PSP |
| IMDB | 3550 | 66428 | 1007 | 3 | MAM |
|  |  | 13788 |  |  | MDM |
| Twitch-Explicit | 9498 | 153138 | 3170 | 2 | DE |
|  | 7126 | 35324 |  |  | ENGB |
|  | 4648 | 59382 |  |  | ES |
|  | 6549 | 1123666 |  |  | FR |
|  | 4385 | 37304 |  |  | RU |
|  | 2772 | 63462 |  |  | TW |

between the decision boundary of classifier and cluster information is weakened. When $p = 1$, $\pi'_{0|D} = \pi_0$ and $\pi'_{1|D} = \pi_1$. In this case, the $Cov(Z, Y) = \mu_D(\pi_1 - \pi_0)$, which has no relations about cluster information. $\square$

# B  More details of Section 4

## B.1  Data statistics

- **Cora** [20]: The Cora is a citation network. The nodes represent papers and are classified into three classes. The edges represent their citation relationships. Node attributes are bag-of-words representations of the papers and the nodes are labeled based on the paper topics.

- **Citeseer** [20]: The Citeseer is a link dataset bulit from citeseer web dataset. The nodes are publications and are divided into six areas. Node attributes are representations of the papers. The edges are citation links.

- **Pubmed** [30]: The Pubmed is a searchable database in the medical field. It consists of nearly twenty thousand nodes. All nodes are divided into three classes. Edges represent papers citation relationship. Node attributes are bag-of-words of the papers.

- **ACM** [27]: This network is extracted from ACM dataset where nodes represent papers and there is an edge between two papers if they have the same author or same subject. So the nodes have two relations which are Papers-Authors-Papers (PAP) and Papers-Subject-Papers (PSP). All the papers are divided into three classes. The features are the bag-of-words representations of paper keywords.

- **IMDB** [27]: IMDB is a movie network dataset where nodes represent movies and there is an edge between two movies if they have the same director or same actor. So the nodes have two relations which are Movie-Actor-Movie (MAM) and Movie-Director-Movie (MDM). All the movies are divided into three classes and features are the bag-of-words of reviews and movie information.

- **Twitch-Explicit** [17]: Twitch datasets contain several networks where nodes represent Twitch users and edges represent their mutual friendships. Each network is collected from a particular region. Different networks have different size, densities and maximum node degrees. All nodes are divided into two classes.

## B.2  Additional results

For more comparison, we show result of deleting edges in Table 4.

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

## B.3 Implementation details

For every GNNs method, we follow the parameter settings from their original paper. SR-GNN and EERM-GNN are initialized with same parameters suggested by their papers and we also further carefully turn parameters to get optimal performance.

For our CIT-GNN, we do not change the parameters of the previous part of GNN backbones and only make an adjustment on our module. Although our transfer process is conducted every $k$ epochs, the clustering process proceeds all the training procedure. For GCN, GAT and GCNII, we put our CIT mechanism before the last layer of GNN. For APPNP, we put it in features extract process, that is, before the last layer of linear transform. We search on the probability of transfer $p$ from 0.05 to 0.3 with step 0.05 and tune epochtimes $k$ of CIT from 2 to 50. For dropout rate, we test ranging is from 0.1 to 0.6. Moreover, we tune the numbers of clusters which is the parameter from spectral clustering from [10, 20, 30, 40, 50, 100, 200]. We set classification loss coefficient, cutloss coefficient and orthogonality loss coefficient as 0.5, 0.3, 0.2 respectively. For all models, we randomly run 5 times and report the average results. For every dataset, we only use original attributes of target nodes, and assign one-hot id vectors to nodes of other types. We report our experiment setting and parameters in supplement.

### B.3.1 Experiment settings

All experiments are conducted with the following setting:

- Operating system: CentOS Linux release 7.6.1810
- CPU: Intel(R) Xeon(R) CPU E5-2620 v4 @ 2.10GHz
- GPU: GeForce RTX 2080 Ti with 11GB and GeForce RTX 3090 with 24GB
- Software versions: Python 3.8; Pytorch 1.10.1; Cuda 11.1;

### B.3.2  Baselines

The publicly available implementations of Baselines can be found at the following URLs:

- GCN: `https://github.com/tkipf/pygcn`
- GAT: `https://github.com/Diego999/pyGAT`
- APPNP: `https://github.com/gasteigerjo/ppnp`
- GCNII: `https://github.com/chennnM/GCNII`
- SR-GNN: `https://github.com/GentleZhu/Shift-Robust-GNNs`
- EERM: `https://github.com/qitianwu/GraphOOD-EERM`

For a fairly comparison, we plug the three methods in same code of GNNs model referred from their papers.

### B.3.3  Hyper parameter settings

Our CIT-GNN contains four hyper-parameter, the probability of transfer $p$, epochtimes $k$, the number of clusters $m$ and $dropout$.

### B.3.4  Settings for Section Perturbation on graph structures data

For Cora, Citeseer and Pubmed, our hyper-parameter settings are as follows respectively:

- CIT-GCN: $p$=0.2/0.1/0.02, $k$=5/5/20,
  $m$=100/20/100, $dropout$=0.5/0.1/0.5 .
- CIT-GAT: $p$=0.1/0.1/0.02, $k$=5/5/5,
  $m$=100/20/100, $dropout$=0.6/0.5/0.3 .
- CIT-APPNP: $p$=0.2/0.2/0.02, $k$=20/20/20,
  $m$=200/10/200, $dropout$=0.6/0.1/0.5 .
- CIT-GCNII: $p$=0.1/0.02/0.1, $k$=5/10/20,
  $m$=100/40/200, $dropout$=0.5/0.3/0.3 .

### B.3.5  Settings for Section Multiplex networks data

For ACM and IMDB (two relations), our hyper-parameter settings are as follows respectively:

- CIT-GCN: $p$=0.2/0.02/0.1/0.05, $k$=10/5/20/5,
  $m$=10/100/40/200, $dropout$=0.5/0.3/0.6/0.3 .
- CIT-GAT: $p$=0.2/0.1/0.2/0.1, $k$=10/5/5/5,
  $m$=10/50/40/50, $dropout$=0.1/0.1/0.1/0.5 .
- CIT-APPNP: $p$=0.2/0.1/0.1/0.1, $k$=5/5/5/5,
  $m$=10/20/40/40, $dropout$=0.5/0.5/0.5/0.3 .
- CIT-GCNII: $p$=0.02/0.1/0.1/0.1, $k$=5/5/5/20,
  $m$=40/100/100/200, $dropout$=0.1/0.3/0.3/0.1 .

### B.3.6  Settings for Section Multigraph data

- CIT-GCN: $p$=0.02, $k$=20, $m$=200, $dropout$=0.3 .
- CIT-GAT: $p$=0.05, $k$=20, $m$=200, $dropout$=0.3 .
- CIT-APPNP: $p$=0.02, $k$=5, $m$=200, $dropout$=0.3 .
- CIT-GCNII: $p$=0.05, $k$=20, $m$=200, $dropout$=0.5 .