# OpenReview forum: "Learning Invariant Representations of Graph Neural Networks via Cluster Generalization"
_NeurIPS.cc/2023/Conference — NeurIPS 2023 poster_

### Official Review · Reviewer_uNQ9 · 2023-07-04

**Soundness:** 2 fair
**Presentation:** 3 good
**Contribution:** 2 fair
**Rating:** 4
**Confidence:** 4

**Summary:**

This paper aims to handle the structure shift problem in GNN. The paper proposed a novel approach -- Cluster Information Transfer (CIT) to address this challenge. CIT first computes cluster center representation during training. CIT then extract the cluster center from the node embedding to retrieve the cluster-independent information. Finally CIT transfers the cluster-independent information to another random cluster center, allowing the model to learn structure-independent knowledges.

**Strengths:**

+ Previous works like EERM, SR-GNN majorly focus on data distribution shifts on graphs, while this paper focus on graph structural shift which is of potential research interest.
+ The method proposed in the paper can be applied on various existing GNN backbones, the method can also fit into different type of tasks and graphs.
+ This paper is well-written and is easy to follow.

**Weaknesses:**

- The CIT method proposed in the paper is not well supported by theorem.  Theoretical analysis only shows that CIT weakens the affect of cluster information. However, the paper still need to answer why it is necessary to transfer nodes across clusters using Eq.8 (If the cluster information is harmful, why not just remove it?) and why can the model learn invariant representations from this process.
- The method introduces a lot of additional hyper-parameters which are sensitive to the results, which are all tuned by grid-search according to the experiments settings. Finally, the accuracy improvements in the experiments is not remarkable compared with other baselines.
- The method is only evaluated on small graphs with several thousand nodes, making it unknown whether the method can be applied to larger graphs.

**Questions:**

Q1：What is the relationship between invariant representations and cluster information and why is transferring a node to another cluster reasonable ?
Q2: Can CIT possibly be scaled to larger graphs?

**Limitations:**

The reviewer does not see potential negative social impact of the work.

---

> ### Author Rebuttal · Authors · 2023-08-10
>
> 1. > W1, Q1：Learning process and theoretical explanation
>
> First, we don't think that cluster information is harmful. The correlation between cluster and labels does help us make predictions. However, this assumption holds when the cluster information does not change, i.e., the structure shifts do not happen. When the structure shifts happen, relying on the original cluster information may lead to mistakes. Therefore, what we need to do is not to remove the cluster information, but learn the cluster-invariant information by simulating cluster changing, so that the model can provide stable predictions under structure shift scenarios. So, to be more precise, we have demonstrated that the CIT mechanism can mitigate the impact of changing clusters during structure shift, enhancing the robustness of the model against such changes. The learning process is similar to the theory of invariant learning [1, 2]. During the process of invariant learning, we train the model to acquire domain-invariant information by generating data of the same sample in different domains. Therefore, similar to our learning process, we employ the CIT mechanism to generate nodes from different clusters to train the model and learn cluster-invariant information.
> We acknowledge that our current explanation may be somewhat unclear and confusing, and we are committed to rectifying these issues in the paper.
>
> 2. > W2：Hyper-parameters analysis and limited accuracy improvement
>
> Our CIT mechanism only consists of three main hyperparameters. $p$ is the probability of transfer. $k$ is the epochtimes of every transfer. $m$ is the number of clusters. As shown in our paper Section 4.4 Analysis of hyper-parameters, the $k$ is consistently stable, and the performance benefits from the applicable selection of $p$ and $m$.
> Our CIT mechanism is a plug-in part that works for different GNN backbone. Therefore, to check the effectiveness of our model, we can see the results of the base model with and without our CIT mechanism. As shown in Table1 and Table2 in our paper,  we conducted significance testing on the experimental results and found that the improvement is significant. For example, in the case of GCNII on Cora dataset, the CIT mechanism improves its accuracy from the original 73.16% to 77.08% with a significance level of 0.01. In the case of APPNP, the CIT mechanism improves its accuracy from the original 75.86% to 78.02% with a significance level of 0.01. Besides, we believe that if we apply CIT to other advanced GNNs and other methods in theory, e.g. SR-GNN, their performance in OOD scenarios can be further improved.
>
> 3. > W3, Q3：Larger graphs
>
>  Our CIT mechanism has two parts of computation: Clustering process and Cluster Information Transfer process. Let $N$ represent the number of nodes, and $K$ represent the number of clusters. The computational complexity of clustering process is $\mathcal{O}(N^2K+NK^2)=\mathcal{O}(NK(N+K))$.  Since the adjacency matrix is usually sparse, the computational complexity can be reduced to $\mathcal{O}(EK)$, where $E$ is the number of non-zero edges in the adjacency matrix. The computational complexity of Cluster Information Transfer process is $\mathcal{O}(pN)$, where $p$ is the probability of transfer. So the computational complexity is $\mathcal{O}(K(E+NK)+pN)$.  The space complexity which depends on the dimension of the assignment matrix is $\mathcal{O}(NK)$.  And we will add this in appendix.
> Our primary contribution lies in proposing a concise and effective CIT mechanism to tackle the structure shift problem, and we have validated its effectiveness.  Actually, it is easy for our model to further lower the computational complexity of the clustering process by replacing spectral clustering with other large-scale graph clustering. For instance, [1] proposes DMoN which uses a new objective function for clustering combining graph spectral modularity maximization and a new regularization method, and the computational complexity of this method is  $\mathcal{O}(d^2N)$. So with the method like this, the computational complexity of our CIT mechanism will reduce to  $\mathcal{O}((d^2+p)N)$, enabling its application on large graphs. However how to extend it to larger graphs is not the focal point of our attention and can be considered as future work.
>
>
> [1] Chang S, Zhang Y, Yu M, et al. Invariant rationalization[C]//International Conference on Machine Learning. PMLR, 2020: 1448-1458.
> [2] Arjovsky M, Bottou L, Gulrajani I, et al. Invariant risk minimization[J]. arXiv preprint arXiv:1907.02893, 2019.
> [3] Müller E. Graph clustering with graph neural networks[J]. Journal of Machine Learning Research, 2023, 24: 1-21.

---

### Official Review · Reviewer_5H2e · 2023-07-04

**Soundness:** 2 fair
**Presentation:** 3 good
**Contribution:** 2 fair
**Rating:** 6
**Confidence:** 4

**Summary:**

The paper introduces a new mechanism called Cluster Information Transfer (CIT) to improve the generalization ability of GNNs to various and unknown test graphs with structure shift. The CIT mechanism enhances the diversity of nodes by combining different cluster information with the nodes, which helps GNNs learn invariant representations. The paper presents experimental results on several benchmark datasets, demonstrating that the proposed CIT-GNN model outperforms existing GNN models in terms of accuracy and macro-F1 score.

**Strengths:**

- The paper is easy to understand.
- While the theory analysis is straightforward, it provides theoretically support for the proposed method.
- The experiments are extensive and the results look promising.


**Weaknesses:**

- **Fair Novelty:** The proposed method is essentially an embedding transformation function that aims to eliminate the statistic variance between clusters that causes bias and hinders generalization. The novelty comes from the clustering mechanism and the augmented training using adjusted node embeddings. However, the spectral clustering mechanism has been introduced previously, and the transformation is relatively straightforward.
- **Assumptions and scope of the solution:** My second concern is about the assumptions of this paper. The paper generally assumes that there exist cluster / structural shifts between training and testing datasets. While structure could change across environments, the node labels remain invariant. The paper also implicitly assumes the node embeddings in different clusters are subject to Multivariate Gaussian distributions. These assumptions should be made explicit and formal. Moreover, the scope of structural shifts needs to be clarified. I understand that the variant of graph density or node degree can be one factor, but do other factors, e.g., edge homophily, also contribute to the shifts? A more general scope should be discussed in order to ensure its applicability.
- **More validation about the claim:** The examples in the introduction considers the shifts between sparse and dense graphs. One simple baseline to mitigate it is to augment the datasets or clusters via DropEdge and AddEdge compared to modifying node embeddings. While I understand that the graph structure is complex in the real world, as the authors claimed (Line 57-59), it would be good to experimentally validate the effectiveness of the method with a simple baseline that modifies the graph structure.
- **Lack of real datasets with natural shifts**: The experiments are somewhat synthetic in the sense that the training and testing sets are splitted with strong structural shifts. I wonder the performance of this method on datasets with more natural shifts, e.g., ogbn-arxiv, where the training and testing sets are splitted based on the time difference.
- **Missing Related Works:** It is unclear how the proposed clustering objective differs from the existing group clustering methods, e.g., Zhou et al. [1].

**Minor:**

- I feel Section 2 is not very informative since the setting is simple and the conclusion is not surprising. It would be good to demonstrate the structural shifts in more real settings or just combine this section to the introduction.
- In Line139 and 149, $m$ and $M$ are both used to indicate the number of clusters. It's better to make it consistent.

In general, the angle of mitigating structure shifts is interesting. However, several things above should be solved or clarified to understand its generality and effectiveness.


[1] Towards Deeper Graph Neural Networks with Differentiable Group Normalization. Kaixiong Zhou, Xiao Huang, Yuening Li, Daochen Zha, Rui Chen, Xia Hu.

**Questions:**

- The method considers the embedding space only. Thus, it seems that the method can be extended to other domains, e.g., for images. If that is the case, are there any graph-specific properties that make this solution remains on the graph domain rather than other general domains?
- What's the method's performance when removing/adding 5%, 20% edges and not removing/adding any edges? Does the improvement of the method decline with the decreasing ratios?

**Limitations:**

The authors discussed the limitations on the fact that the method could only apply to node-level tasks directly and the theory remain to be fully developed, as well as the limitation that the transfer remains on embedding space but graph topology.

---

> ### Author Rebuttal · Authors · 2023-08-10
>
> 1. >W1, Q1： Fair Novelty
>
> We agree that the spectral clustering mechanism has been introduced previously, however, in this work, our main contribution resides in the development of the CIT mechanism and spectral clustering is just one component of it. The CIT mechanism is a flexible and succinct approach compatible with different GNN backbones, which has not been discussed in this way at present. Despite our CIT mechanism is relatively simple, we think that "simple yet effective" is actually our final goal in practice.
>
> Despite our model considers the embedding space only, it is actually technically challenging to reasonably model domain information. First, given graph structure, how to capture the domain information of a node is not well explored. Usually, for images, we can consider the background information as the domain information [8]. For graph data, we just know the structure which lacks semantic information and is very unintuitive. Here, we initially consider utilizing clustering information as local information. Given the message-passing mechanism in GNNs, the local properties of nodes are determinants of representation learning. Therefore, a structure shift might correlate to a shift in these local properties of a node. Coincidentally, the cluster information typically encapsulates the local attributes of nodes and can be harnessed to generate various local properties.
>
> 2. > W2： Scope of the solution
>
>  With regard to structure shift in graph data, there is a prevalent discourse surrounding out-of-distribution (OOD) graph classification, such as molecular scaffold [2, 3]. Conversely, there is relatively less discussion regarding node classification and it needs to be explored. In the context of a node classification task, we think that local properties of nodes, including neighbor information under the message-passing mechanism, are crucial determinants of node representation learning. Therefore, a structure shift might correlate to a shift in these local properties of a node, which could manifest as variations in node degree, edge density, graph size, or homophily.
>
> 3. > W2：Assumptions
>
> The first assumption you said is our structure shift problem setting.
> Regarding the second assumption, we follow [4,5] and consider the structure as the environment.   Meanwhile, referencing the theory of invariant learning [6], we learn invariant representations by maintaining label invariant across different structures.
> As for the third assumption pertaining to Gaussian distribution, it's important to clarify that we do not make this assumption in our model. In Section 2, we employ multivariate Gaussian distribution to generate nodes' features to establish a binary classification setting, which is a very common operation [7], facilitating the observation of experimental phenomena. Meanwhile, we introduce Gaussian perturbations to enhance the diversity of cluster information in embedding space. Therefore, we do not posit that the cluster distribution is subject to Multivariate Gaussian distributions.
>
> 4. > W3：Simple baselines
>
> Thanks for the suggestion. We conducted experiments involving the use of the DropEdge and AddEdge techniques on the GNN backbones we employed, and the experimental results are shown in global response Table 2.
> The experimental results indicate that even simple techniques like AddEdge and DropEdge can enhance the performance of the original GNN. This also validates the correctness of our approach to learn invariant information by altering node domains to address structure shift.
>
> 5. > W4：Natural shifts
>
> Actually we applied our model to real datasets with natural shifts. For Multiplex networks, we select two datasets, ACM and IMDB, which have two relation structures, and we train on one relation structure and test on another. For example, in the case of the ACM network, we train on PAP relation where edges represent that two papers have the same author and test on PSP relation where edges represent that two papers belong to the same subject. All these relations naturally exist.
> For Multigraph data, we follow [4] and select Twitch-Explict dataset. Twitch datasets contain several networks and each network is collected from a particular region. Therefore these are natural shift.
>
> 6. > W5：Related Works
>
> Firstly, the purpose of the clustering process between [1] and ours is different.
> [1] employs clustering in order to group nodes with consistent labels and utilizes cluster information to alleviate over-smoothing issues on the graph. During this process, node labels will be utilized for clustering. However, our clustering aims to group similar nodes to model local properties through clustering information, without leveraging label information.
> Secondly, our technical approach is also different. Our work utilizes spectral clustering and the purpose of operating in the embedding space is to transform clusters. [1] achieves clustering by optimizing two distances and normalizing cluster representation in the embedding space. And we will cite it.
>
> 7. > Q2：Improvement of the method decline with the decreasing ratios
>
> We conducted experiments and the results are shown in global response Table 3, and we can see the improvement becomes less pronounced as the magnitude of perturbation decreases.
>
> [2] Li H, Zhang Z, Wang X, et al. Learning invariant graph representations for out-of-distribution generalization[J].
>
> [3] Miao S, Liu M, Li P. Interpretable and generalizable graph learning via stochastic attention mechanism[C].
>
> [4] Wu Q, Zhang H, Yan J, et al. Handling Distribution Shifts on Graphs: An Invariance Perspective[C].
>
> [5] Zhang Z, Wang X, Zhang Z, et al. Dynamic graph neural networks under spatio-temporal distribution shift[J].
>
> [6] Chang S, Zhang Y, Yu M, et al. Invariant rationalization[C].
>
> [7] Wang X, Zhu M, Bo D, et al. Am-gcn: Adaptive multi-channel graph convolutional networks[C].
>
> [8] Zhou K, Yang Y, Qiao Y, et al. Domain generalization with mixstyle[J].

---

> > ### Comment · Reviewer_5H2e · 2023-08-12
> > **Re author rebuttal**
> >
> > Thanks for the authors's response which solved part of my concerns.
> >
> > One concern that hasn't been addressed is that, the scope of the solution is based on the assumption of the cluster structure shifts. And I feel the response "Therefore, a structure shift might correlate to a shift in these local properties of a node, which could manifest as variations in node degree, edge density, graph size, or homophily" is not accurate, how does looking into different clusters of nodes mitigates the distribution shifts on graph size (which is global information)? Still, I prefer the author to make the assumptions more explicit and discuss how common the cluster structure shifts happen in the real-world graph datasets.
> >
> > Secondly, I think the authors may misunderstand my concern on Related Works. I wasn't asking about the difference between the specific clustering method and your method, but in a more general context - graph clustering methods, since it's also a crucial componant in your framework and could largely affect the performance.

---

> > > ### Author Response · Authors · 2023-08-14
> > >
> > > We sincerely thank the Reviewer for the valuable opinions and concerns about our work. To address your concerns, we will give more explicit explanations.
> > >
> > > Here, we think that the structure shift is highly related with the local structure shift of nodes, resulting in the changes in information aggregated by nodes. For graph data, cluster information captures information from similar and connected nodes, closely related to the local properties of nodes. Meanwhile, cluster information is determined by both features and structure. When there are changes in the structure, the cluster information also changes. Thus, we utilize cluster information to model the local properties of nodes, and characterize changes in the structure through variations in clusters.   And in the real world, it is also common for cluster information to change intuitively. For instance, in adversarial attack scenarios, manipulating the edges of targeted nodes can result in changes to node cluster information. Similarly, in social networks, the cluster information of users can shift based on diverse social behaviors and interactions, among other factors.
> > > As for the issue of graph size, what we want to convey is that changes in the local information of nodes may be reflected in the variation of the number of nodes,  such as an increase or decrease in neighbors of the node. In this scenario, the change in the graph size is still caused by the change in the local information of nodes. Therefore, although we didn't model the graph size information explicitly, we can still enhance the performance of model in cases with shifts in graph size by making the model more robust to changes in local information. Our experiment (for details, please refer to Section 4.3) also includes this case. We train the model on the graph with 6549 nodes, while the testing graphs have both more and fewer nodes than it, and experimental results demonstrate that we have improved the performance of the models. We sincerely thank the Reviewer for spending time and providing valuable feedback, and we will make it more explicit in our paper.
> > >
> > > We are really sorry for misunderstanding your concerns about the Related Works.  For the clustering part, we employed an existing spectral clustering method that falls within the scope of graph clustering methods. But the clustering method used here needs to be differentiable and capable of preventing cluster collapse to integrate with our framework. Technically, the clustering component could be substituted with other types of clustering methods to achieve different effects, which is well worth exploring and can be considered as future work. Thank you very much for your suggestions; we will add the related work on graph clustering in our paper.

---

> > > > ### Comment · Reviewer_5H2e · 2023-08-14
> > > > **Response**
> > > >
> > > > I appreciate the authors' response.
> > > >
> > > > I agree the point that it's challenging to reasonably model domain information of graphs. The assumptions still seem a bit blurry to me but the examples make sense to me. I also read other reviewers' comments and I think it's true that the hyperparameters are a bit sensitive.
> > > >
> > > > I am willing to raise my score to weak accept given some of my concerns are solved.

---

> > > > > ### Author Response · Authors · 2023-08-15
> > > > >
> > > > > We sincerely appreciate your response! And thanks again for your time and efforts in reviewing our work.
> > > > >
> > > > > Best regards.

---

### Official Review · Reviewer_Nbx4 · 2023-07-05

**Soundness:** 3 good
**Presentation:** 3 good
**Contribution:** 4 excellent
**Rating:** 7
**Confidence:** 4

**Summary:**

This paper focuses on the invariant learning of graph neural networks and proposes a cluster information transfer mechanism with two statistics: the mean of cluster and the variance of cluster. The authors prove that with CIT mechanism, the model is able to capture the cluster independent information, so as to improve the generalization ability across different structure shifts. Experiments on different graph benchmarks show promising results of CIT mechanism.

**Strengths:**

Introduction and background parts are clearly written and motivate this study well. The research problem is important, the ideas are clearly clarified and all the technical steps are easy to follow. The paper provides the comprehensive experiments on different kinds of graphs to evaluate the performance of the proposed model.


**Weaknesses:**

No specific discussions on structure shift. Time complexity analysis is not provided. The CIT mechanism is not explained sufficiently.

**Questions:**

1. It seems that most of the graphs used here are not large enough, so what is the time complexity of the proposed model?
3. The authors design several experiments to evaluate the model, e.g., perturbate edges, multiplex graphs. Do all these changes belong to the structure shift? Can the authors provide more discussions on what changes can be considered as structure shift?
4. The paper proves that with CIT mechanism, the cluster information has less influence on the classifier, but what if the cluster information is related with the labels?

---

> ### Author Rebuttal · Authors · 2023-08-10
>
> 1. > Q1：The time complexity of the proposed model
>
>  Our CIT mechanism has two parts of computation: Clustering process and Cluster Information Transfer process. Let $N$ represent the number of nodes, and $K$ represent the number of clusters. The computational complexity of clustering process is $\mathcal{O}(N^2K+NK^2)=\mathcal{O}(NK(N+K))$.  Since the adjacency matrix is usually sparse, the computational complexity can be reduced to $\mathcal{O}(EK)$, where $E$ is the number of non-zero edges in the adjacency matrix. The computational complexity of Cluster Information Transfer process is $\mathcal{O}(pN)$, where $p$ is the probability of transfer. So the computational complexity is $\mathcal{O}(K(E+NK)+pN)$.  The space complexity which depends on the dimension of the assignment matrix is $\mathcal{O}(NK)$.  And we will add this in appendix.
>
> Our primary contribution lies in proposing a concise and effective CIT mechanism to tackle the structure shift problem, and we have validated its effectiveness.  Actually, it is easy for our model to further lower the computational complexity of the clustering process by replacing spectral clustering with other large-scale graph clustering. For instance, [1] proposes DMoN which uses a new objective function for clustering combining graph spectral modularity maximization and a new regularization method, and the computational complexity of this method is  $\mathcal{O}(d^2N)$. So with the method like this, the computational complexity of our CIT mechanism will reduce to  $\mathcal{O}((d^2+p)N)$, enabling its application on large graphs. Therefore how to extend it to larger graphs is not the focal point of our attention and can be considered as future work.
>
> 2. > Q2：Discussions about structure shift
>
>  According to [2,3],  test graphs differ in numerous structural aspects - such as size, edge density, and degrees - compared to their training graphs, which can be seen as structure shifts. We focus on three primary scenarios: perturbations in graph structures, Multiplex networks, and Multigraph data. In the case of perturbations in graph structures, random edge additions and deletions are implemented. For multiplex networks, varied relationships among nodes are considered. As for multigraph data, we utilize graphs drawn from diverse regions.  A comprehensive statistical analysis of structure shifts in graph data and label rates is shown in global response Table 1. It can be observed that the experiments we have set up are all based on structure shift.
>
>  With regard to structure shift in graph data, there is a prevalent discourse surrounding out-of-distribution (OOD) graph classification, such as molecular scaffold [4, 5]. Conversely, node classification discussions remain relatively sparse and need to be explored. In the context of a node classification task, we think that local properties of nodes, including neighbor information under the message-passing mechanism, are crucial determinants of nodes representation learning. Therefore, a structure shift might correlate to a shift in these local properties of a node, which could manifest as variations in node degree, edge density, graph size, or homophily.
>
> 1. > Q3：What if the cluster information is related with the labels?
>
>  In this work, we aim at learning the cluster-invariant information with CIT mechanism. Therefore, when the structure shifts happen, the model can make stable predictions under different structures. As for the correlation between the cluster information and the labels, this assumption may work well with no distribution shifts, however, if the distribution shifts happen, which is the scenario that we focus on in this paper, the assumption may not hold.
> In this situation, the CIT mechanism simulates structure shift by altering the cluster information of nodes, allowing nodes to exist in different domains.  So the model learns cluster-invariant information by training on nodes from multiple clusters. So, to be more precise, we have demonstrated that the CIT mechanism can mitigate the impact of changing clusters during structure shift, enhancing the robustness of the model  against such changes.
>
>  [1] Müller E. Graph clustering with graph neural networks[J]. Journal of Machine Learning Research, 2023, 24: 1-21.
> [2] Wu Q, Zhang H, Yan J, et al. Handling Distribution Shifts on Graphs: An Invariance Perspective[C]//International Conference on Learning Representations. 2021.
> [3] Wu Q, Chen Y, Yang C, et al. Energy-based Out-of-Distribution Detection for Graph Neural Networks[C]//The Eleventh International Conference on Learning Representations. 2022.
> [4] Li H, Zhang Z, Wang X, et al. Learning invariant graph representations for out-of-distribution generalization[J]. Advances in Neural Information Processing Systems, 2022, 35: 11828-11841.
> [5] Miao S, Liu M, Li P. Interpretable and generalizable graph learning via stochastic attention mechanism[C]//International Conference on Machine Learning. PMLR, 2022: 15524-15543.

---

> > ### Comment · Reviewer_Nbx4 · 2023-08-19
> > **Thanks for your response.**
> >
> > Thanks for your response. I have no further questions.

---

> > > ### Author Response · Authors · 2023-08-20
> > >
> > > We sincerely appreciate your feedback and acknowledgement of our rebuttal. And thanks again for your time and efforts in reviewing our work.
> > >
> > > Best regards.

---

### Official Review · Reviewer_nmEc · 2023-07-06

**Soundness:** 3 good
**Presentation:** 3 good
**Contribution:** 4 excellent
**Rating:** 7
**Confidence:** 5

**Summary:**

The paper tackles an important question of learning invariant representations of GNNs. The authors show that once the test graph pattern shifts, the reliability of GNNs becomes compromised. Then they propose the cluster information transfer mechanism, which can be easily combined with current GNNs to improve their robustness on structure shift. The authors present numerical experiments that suggest that the proposed mechanism is effective.

**Strengths:**

1.	I enjoyed reading the paper. The presentation is clear and mostly easy to follow.

2.	Clear results on different scenarios, well-backed by experiments.

3.	The CIT mechanism is interesting and with technical analysis.


**Weaknesses:**

1.	In section 2, why GAT performs worse than GCN?

2.	In fig.2, the model uses the node representation learned by GNNs to obtain clusters, so does the representation learned by different layers affect the clustering process?

3.	The authors mentioned that the graph OOD benchmark is built, so why the datasets used in the experiments are still the traditional graphs?


**Questions:**

It would be beneficial to answer the questions in weakness.

---

> ### Author Rebuttal · Authors · 2023-08-10
>
> 1. > W1： In section 2, GAT performs worse than GCN
>
> We consider that the attention mechanism of GAT is a potential catalyst for the observed phenomenon. Existing research [1, 2] suggests that the attention mechanism is easily influenced by the distribution of neighbor characteristics. In Section 2, we design experiments to demonstrate the effect of structure shift on GNN performance. We set up two classes and generate features through Multivariate Gaussian distributions. In generating the test set graph structure, we changed the probability of edges between inter-class and intra-class. In this case, the homophily of node neighbors decreases, and heterophily increases. Thus the parameters of GAT learned based on the attention of the original neighbors are no longer applicable to the new neighbors, which exacerbates the decline in GAT's performance.  However, GCN does not have this attention mechanism, so in this scenario, the parameters learned by GCN are relatively more robust, resulting in a less drastic performance decline.
>
> 2. > W2：Does the representation learned by different layers affect the clustering process?
>
> We believe that the representations learned by different layers affect cluster processing.  Given the message-passing mechanism on GNNs, multilayer networks are capable of aggregating information from multiple top neighbors.
> So theoretically, the representations learned from deep layers will aggregate more information and also be more conducive to the clustering process. However, in the real scenarios, due to the impact of over-smoothing, GNNs like GCN and GAT often consist of only two layers. Moreover, in GNNs, the final layer frequently serves as a classifier. Therefore, we employ the representations learned from the first layer for clustering. For networks that can be deeper, such as GCNII, we utilize representations learned before the final layer for clustering purposes.
>
> 3. > W3： Why the datasets used in the experiments are still the traditional graphs?
>
> The traditional graphs we used in our work are also used in Good benchmark [3], with the distinction that the data has been partitioned based on various out-of-distribution problems.  Meanwhile, our work focuses on structure shift poblems that differ from Good benchmark.  Good presents OOD problem according to two distinct formulations of distribution shift, namely covariate shift and concept shift,  and data sampling and partitioning are performed based on these two shifts, which is not suitable for our problem. What we focus on is structure shift, and data partitioning is done based on the differences in structure.
>
> [1] Veličković P, Cucurull G, Casanova A, et al. Graph attention networks[J]. arXiv preprint arXiv:1710.10903, 2017.
> [2] Brody S, Alon U, Yahav E. How attentive are graph attention networks?[J]. arXiv preprint arXiv:2105.14491, 2021.
> [3] Gui S, Li X, Wang L, et al. Good: A graph out-of-distribution benchmark[J]. Advances in Neural Information Processing Systems, 2022, 35: 2059-2073.

---

### Official Review · Reviewer_QyZE · 2023-07-07

**Soundness:** 4 excellent
**Presentation:** 3 good
**Contribution:** 4 excellent
**Rating:** 7
**Confidence:** 5

**Summary:**

The paper introduces a novel Cluster Information Transfer (CIT) mechanism to enhance the generalization capability of Graph Neural Networks (GNNs) in the presence of structural shifts. The authors provide theoretical analysis, showing that the impact of cluster information on the classifier diminishes during transfer. They demonstrate that the CIT mechanism can be easily incorporated into existing models and showcase its effectiveness through extensive experiments. Overall, the paper highlights the value of the CIT mechanism in improving GNN generalization and supports its claims with comprehensive empirical evidence.

**Strengths:**

- The paper includes both theoretical analysis and numerical results, providing a well-rounded evaluation of the proposed approach.
- The idea presented in the paper is clear and well-motivated, addressing an important problem.
- The experiments conducted in the paper are extensive and provide convincing evidence to support the claims made by the authors.

**Weaknesses:**

- The experiments in section 2 are not sufficient.
- The descriptions of the experimental settings are unclear and could benefit from more detailed and precise explanations.
- There are some typos present in the paper that should be addressed for improved clarity and readability.

**Questions:**

- In Section 2, the authors focus on presenting results specifically for the GCN and GAT models, but they do not mention the evaluation of other GNN models. It would be beneficial for the authors to clarify if they have considered or conducted experiments with other GNN models as well.
- The authors introduce different structure shifts in their experiments, but it would be helpful to provide a clearer distinction between their approach and the OOD graph benchmarks proposed in [1]. Additionally, the paper could benefit from providing standard settings for structure shifts and including information on the label rate used in training the models.
- There are several typos in the paper, such as "less affect" in line 321, "bulids" in line 315, and "Initial residual and Identity mapping two effective techniques on GCN" in line 305. These errors should be addressed to improve the clarity and accuracy of the paper.

[1] Shurui Gui, Xiner Li, Limei Wang, and Shuiwang Ji. Good: A graph out-of-distribution benchmark. arXiv preprint arXiv:2206.08452, 2022.

**Limitations:**

The authors have discussed the limitation and potential negative societal impact.

---

> ### Author Rebuttal · Authors · 2023-08-09
>
> 1. > Q1：More GNN results
>
> Thanks for the suggestion. We evaluate another two popular GNNs (APPNP [2] and GCNII [3]), where the results are shown as follows.
>
> |       | 0.5 0.05 | 0.45 0.1 | 0.4 0.15 | 0.35 0.2 | 0.3 0.25 | 0.25 0.3 |
> |-------|----------|----------|----------|----------|----------|----------|
> | GCN   | 97.37    | 92.34    | 90.1     | 86.45    | 81.04    | 76.88    |
> | GAT   | 97.41    | 92.08    | 88.33    | 81.77    | 76.59    | 69.58    |
> | APPNP | 97.8     | 93.22    | 90.21    | 89.58    | 83.33    | 80.72    |
> | GCNII | 98.54    | 95.41    | 91.39    | 87.89    | 84.2     | 81.35    |
>
>  The numbers in the table headers represent the edge generation probabilities for inter-community and intra-community, and the experimental setup remains consistent with that described in Section 2 (for details, please refer to Section 2). Building upon the original foundation, we conducted experiments involving APPNP and GCNII. From the experimental results, it can be observed that with the changes in structure, the performance of these GNNs has experienced varying degrees of decline, which shows that once the test graph pattern shifts, the reliability of GNNs becomes compromised. We will show them together in our paper.
>
> 2. > Q2：Difference with Good
>
> While both our work and the Good benchmark [1] address graph OOD problems, the reasons for causing the distribution shift that we focus on are distinct, which leads to differences in our data partitioning as well. Good presents OOD problem according to two distinct formulations of distribution shift, namely covariate shift and concept shift. The former refers to scenarios where the input distributions exhibit variance between training and testing data, formally described as $P_{train}(X) \ne P_{test}(X), P_{train}(Y|X) = P_{test}(Y|X)$. In contrast, the concept shift refers to instances where the conditional distribution experiences variation, expressed formally as $P_{train}(X) = P_{test}(X), P_{train}(Y|X) \ne P_{test}(Y|X)$.  Thus the data sampling and partitioning of Good are performed based on these two shifts. The investigation in our paper concentrates on the unique OOD phenomenon triggered by graph structure shifts which are exclusive to graph data formalized as $P_{train}(A) \ne P_{test}(A)$. This shift opens up two potential conditional distribution: $P_{train}(Y|X, A) = P_{test}(Y|X, A), P_{train}(Y|X, A) \ne P_{test}(Y|X, A)$. What we focus on is structure shift, and data partitioning is done based on the differences in structure. Therefore, the data partitioning of Good is not applicable to our problem.
>
> 3. >Q2：Standard settings and label rate
>
>  According to [4, 5],  test graphs differ in numerous structural aspects - such as size, edge density, and degrees - compared to their training graphs, which can be seen as structure shifts. We focus on three primary scenarios: perturbations in graph structures, Multiplex networks, and Multigraph data. In the case of perturbations in graph structures, random edge additions and deletions are implemented. For multiplex networks, varied relationships among nodes are considered. As for multigraph data, we utilize graphs drawn from diverse regions.
> Data partitioning for cora, citeseer, and pubmed follows the methodology detailed in [6], while for ACM and IMDB, we follow the approach presented in [7]. As for Twitch-Explicit, we base our data division on the methods proposed in [4]. A comprehensive statistical analysis of structure shifts in graph data and label rates is included in global response Table 1.
>
> 4. >Q3：Typos
>
> Thanks for pointing this out, and we will correct these typos.
>
> [1] Gui S, Li X, Wang L, et al. Good: A graph out-of-distribution benchmark[J]. Advances in Neural Information Processing Systems, 2022, 35: 2059-2073.
> [2] Gasteiger J, Bojchevski A, Günnemann S. Predict then propagate: Graph neural networks meet personalized pagerank[J]. arXiv preprint arXiv:1810.05997, 2018.
> [3] Chen M, Wei Z, Huang Z, et al. Simple and deep graph convolutional networks[C]//International conference on machine learning. PMLR, 2020: 1725-1735.
> [4] Wu Q, Zhang H, Yan J, et al. Handling Distribution Shifts on Graphs: An Invariance Perspective[C]//International Conference on Learning Representations. 2021.
> [5] Wu Q, Chen Y, Yang C, et al. Energy-based Out-of-Distribution Detection for Graph Neural Networks[C]//The Eleventh International Conference on Learning Representations. 2022.
> [6] Kipf T N, Welling M. Semi-supervised classification with graph convolutional networks[J]. arXiv preprint arXiv:1609.02907, 2016.
> [7] Wang X, Ji H, Shi C, et al. Heterogeneous graph attention network[C]//The world wide web conference. 2019: 2022-2032.

---

### Author Rebuttal · Authors · 2023-08-10

We sincerely thank all the reviewers for your careful reading and suggestions. Due to character constraints, we have uploaded the experimental results in this PDF.

---

### Decision · Program_Chairs · 2023-09-21

**Decision:**

Accept (poster)

**Comment:**

The paper introduces a new mechanism called Cluster Information Transfer (CIT) to improve the generalization ability of GNNs in the presence of structure shifts. The CIT mechanism enhances the diversity of nodes and helps learn invariant representations by combining different cluster information with the nodes. The CIT mechanism can be easily incorporated into existing models. A theoretical analysis is presented, showing that the impact of cluster information on the classifier diminishes during transfer.  Overall the paper proposes a practical solution that addresses an important problem.